# Encapsulation of Nedaplatin in Novel PEGylated Liposomes Increases Its Cytotoxicity and Genotoxicity against A549 and U2OS Human Cancer Cells

**DOI:** 10.3390/pharmaceutics12090863

**Published:** 2020-09-10

**Authors:** Salma El-Shafie, Sherif Ashraf Fahmy, Laila Ziko, Nada Elzahed, Tamer Shoeib, Andreas Kakarougkas

**Affiliations:** 1Department of Biology, School of Sciences and Engineering, The American University in Cairo, Cairo 11835, Egypt; salma4@aucegypt.edu (S.E.-S.); laila.adel@aucegypt.edu (L.Z.); nada.elzahed@aucegypt.edu (N.E.); 2Department of Chemistry, School of Sciences and Engineering, The American University in Cairo, Cairo 11835 Egypt; sheriffahmy@aucegypt.edu

**Keywords:** chemotherapeutics, liposomes, platinum drugs, nedaplatin, DNA repair, cancer treatment

## Abstract

Following the discovery of cisplatin over 50 years ago, platinum-based drugs have been a widely used and effective form of cancer therapy, primarily causing cell death by inducing DNA damage and triggering apoptosis. However, the dose-limiting toxicity of these drugs has led to the development of second and third generation platinum-based drugs that maintain the cytotoxicity of cisplatin but have a more acceptable side-effect profile. In addition to the creation of new analogs, tumor delivery systems such as liposome encapsulated platinum drugs have been developed and are currently in clinical trials. In this study, we have created the first PEGylated liposomal form of nedaplatin using thin film hydration. Nedaplatin, the main focus of this study, has been exclusively used in Japan for the treatment of non-small cell lung cancer, head and neck, esophageal, bladder, ovarian and cervical cancer. Here, we investigate the cytotoxic and genotoxic effects of free and liposomal nedaplatin on the human non-small cell lung cancer cell line A549 and human osteosarcoma cell line U2OS. We use a variety of assays including ICP MS and the highly sensitive histone H2AX assay to assess drug internalization and to quantify DNA damage induction. Strikingly, we show that by encapsulating nedaplatin in PEGylated liposomes, the platinum uptake cytotoxicity and genotoxicity of nedaplatin was significantly enhanced in both cancer cell lines. Moreover, the enhanced platinum uptake as well as the cytotoxic/antiproliferative effect of liposomal nedaplatin appears to be selective to cancer cells as it was not observed on two noncancer cell lines. This is the first study to develop PEGylated liposomal nedaplatin and to demonstrate the superior cell delivery potential of this product.

## 1. Introduction

It is now over 50 years since the accidental discovery of cisplatin as an antitumor agent by the Rosenberg laboratory at Michigan State University. To this day, cisplatin and other platinum-based drugs form the backbone of cancer treatment, with more than 50% of cancer patients receiving chemotherapy using platinum drugs [1]. Cisplatin was originally approved for use in testicular and ovarian cancers and has been used worldwide to treat various types of cancers including sarcomas, carcinomas, lymphomas, cervical cancer, bladder cancer and germ cell tumors [2,3].

Platinum-based drugs exert their antitumor effect by binding to DNA and forming DNA adducts. Distortions in DNA caused by platinum-DNA adducts disrupt cellular processes and activate the DNA damage responses that include cell cycle arrest, DNA repair and apoptosis [4,5]. Cisplatin shows remarkable antitumor activity but its use in cancer treatment is limited due to side effects that include severe nephrotoxicity and gastrointestinal disorders. In addition, some tumors are inherently resistant to cisplatin whilst others acquire resistance via mechanisms resulting in reduced DNA binding or via up regulation of the DNA damage responses leading to increased cell survival [6,7].

The dose limiting toxicity of cisplatin as well as the problem of innate and acquired resistance drove the effort for the discovery of more effective and less toxic analogs. Second generation platinum-based drugs include carboplatin, which maintains the cytotoxic effect of cisplatin but does not display nephrotoxicity, although other dose-limiting side effects are present [8,9]. Third generation drugs include oxaliplatin, which was developed to overcome cisplatin resistance. The uptake of oxaliplatin is less dependent on the CTR1 copper transporter, whilst the formed platinum-DNA adducts are not efficiently repaired by the DNA repair machinery leading to a better response in cancers that have acquired resistance to cisplatin [10].

Of the thousands of other cisplatin analogs that have been synthesized and tested, only a very small number have received FDA approval and are in clinical use [3]. In the past couple of decades, research has shifted towards ways of improving the efficacy of existing drugs as this can help identify promising combinations for further clinical development. Of particular relevance to this study, research into nanocarrier-based delivery to tumor cells is greatly expanding because of its potential in improving drug efficacy, reduction of side effects and overcoming drug resistance [3,9,11]. 

Liposomes are an example of a drug delivery system with great promise in cancer treatment. Liposomes are spherical vesicles with an aqueous inner core surrounded by one or more concentric bilayers of phospholipids. Since first being engineered in the 1960s, the physical properties of liposomes have been modified to increase their efficacy as drug carriers. Moreover, the introduction of PEGylated liposomes, known as stealth or long-circulating liposomes, triggered-release and ligand-targeted liposomes, have further increased the potential of this drug delivery system [12].

A total of six platinum-based liposomal drugs are currently in clinical trials; lipoplatin, SPI-077 and LiPlaCis, all encapsulating cisplatin, are in clinical trial phases II/III, II and I, respectively. Lipoxal and MBP-426, encapsulating oxaliplatin, are in phases I and I/II, respectively and finally Aroplatin, encapsulating NDDP, a platinum analog structurally similar to oxaliplatin has completed phase II trials [1]. Lipoplatin, a PEGylated liposome, has shown advantages such as long-term circulation of cisplatin, 200-fold higher accumulation of cisplatin in cancer tissues compared to adjacent normal tissue, considerably reduced toxicity, and increased ability to fuse and penetrate the cell membrane [13].

In this study we have synthesized a novel formulation of PEGylated liposomal nedaplatin (LND). Nedaplatin is a second-generation platinum-based drug that has been exclusively used in Japan since 1995 for the treatment of non-small cell lung cancer, head and neck, esophageal, bladder, ovarian and cervical cancer [1]. Nedaplatin, like carboplatin, has lower nephrotoxicity than cisplatin but leads to platinum-DNA adducts identical to those caused by cisplatin [11]. Moreover, recent clinical trials and meta-analyses comparing nedaplatin to cisplatin showed noninferior efficacy of nedaplatin and even improved health-related quality of life as it led to significantly less adverse events such as nausea, vomiting, ototoxicity, kidney toxicity compared to cisplatin [14,15,16,17,18,19]. However, nedaplatin leads to thrombocytopenia caused by marrow suppression and this toxicity is usually reported at the drug therapeutic dose, leading to alteration of the chemotherapy regimen or discontinuation [20]. Here we report that by performing liposomal encapsulation of nedaplatin we were able to increase the cytotoxicity and genotoxicity of nedaplatin on cultured human cancer cells through increased cellular uptake of platinum. This study therefore identifies liposomal nedaplatin (LND) as a promising candidate for future clinical development.

## 2. Methods

### 2.1. Reagents

Nedaplatin was supplied by Shandong Boyuan Pharmaceutical Co., Ltd. (Shandong, China). DSPC, DSPE, MPEG-2000-DSPE and cholesterol were purchased from Corden Pharma (Plankstdt, Germany). HPLC-grade acetonitrile and dichloromethane (DCM), were purchased from Sigma-Aldrich (Hamburg Germany). Milli-Q ultrapure water was provided using the Millipore system (Watford, UK). For the biological assays all cell culture reagents were purchased from Lonza (Bornem, Belgium), 1-(4,5-Dimethylthiazol-2-yl)-3,5-diphenylformazan (MTT) was purchased from Serva (Heidelberg, Germany), Cytochalasin B and 4′,6-diamidino-2-phenylindole (DAPI) nuclear stain from Sigma-Aldrich (Hamburg, Germany).

### 2.2. Liposome Preparation

DSPC/DSPE/MPEG-2000-DSPE/cholesterol (1:0.1:0.1:0.8) liposomes were prepared by thin-film hydration as previously described [21]. Briefly, the lipid composition (0.0412 mmole lipids and cholesterol) was dissolved in DCM in a round bottom flask. Then the organic solvent was removed under vacuum in a rotary evaporator operating on 150 rpm at 60 °C, which nearly equals the transition temperature of the lipids, to form a thin lipid film. This film was rehydrated in 3 mL of 7.55 mM nedaplatin solution, prepared in phosphate buffer saline (PBS) of pH 7.4, at 60 °C for two hours (at a 1:2 drug: lipid molar ratio). The resulting multivesicular liposomes were extruded for 15 cycles through 0.1 µm polycarbonate membranes mounted in an Avanti Mini extruder (Avanti polar lipids Inc., Alabama, AL, USA) to produce samples with a narrow size distribution. The extrusion was carried out at 60 °C to maintain vesicles above phase transition temperature. The liposomal formulations were stored at 4 °C until use. Void liposomes used as control were prepared by the same method with appropriate volume of PBS (pH 7.4) being used during rehydration instead of the nedaplatin solution.

### 2.3. Liposome Characterization

#### 2.3.1. Entrapment Efficiency

The prepared liposomes were characterized with respect to their entrapment efficiency (EE), size distribution, zeta potential, stability, morphology and in vitro release kinetics. Entrapment efficiency, defined as the encapsulated and surface immobilized nedaplatin, was determined by the centrifugation of the liposomal sample at 16,500 rpm at 4 °C for 3 h, after which the supernatant containing the non-entrapped aqueous Nedaplatin (ND) was separated from the surface, immobilized and the liposome entrapped ND in the pellet residue. Aliquots from supernatant and pellet residue were collected. The pellet residue collected was diluted 1:4 (*v*/*v*) in 19.28 mM SDS solution for complete liposome disruption and subsequent release of the entrapped ND. Quantitative determination of ND was performed by HPLC (Appendix A). The entrapment efficiency, defined as the sum of encapsulated and surface entrapped drugs, expressed in percentage (%), were calculated using the following equation:$$Entrapment\;efficiency\;\%=\frac{Total\;amount\;of\;drug\;entrapped}{Total\;amount\;of\;drug}\times 100$$

#### 2.3.2. Particle Size and Zeta Potential

The hydrodynamic diameter, polydispersity and zeta potential of liposomes were determined by the dynamic light scattering technique utilizing a Zetasizer Nano ZS (Malvern Instruments, Co., Malverny, UK) equipped with a 10 mW HeNe laser. All samples were diluted to 1:10 (*v*/*v*) with ultra-pure water and measured at 25 °C in triplicates at a wavelength of 633 nm and a detection backscatter angle of 173°.

#### 2.3.3. Morphology

The morphology of the liposomal formulation was examined using transmission electron microscopy (TEM) employing a JEOL-JEM 2100 electron microscope operating at 160 kV. A 50 µL aliquot of a liposomal sample diluted to 1:2 (*v*/*v*) with ultra-pure water was stained with 2% aqueous phosphotungstic acid. This mixture was deposited and dried over a carbon-coated copper 200 mesh grid, visualized and photographed.

#### 2.3.4. In Vitro Drug Release

The in vitro drug release profiles of ND encapsulated in stealth liposome formulation was determined using dialysis for separating the nonentrapped drug from the liposomes as the molecular weight cutoff of dialysis tubing is 12KDa. The release study was conducted in PBS and fetal bovine serum (FBS) media at pH 7 and 6.5, respectively. Liposomal solution (0.5 mL) was placed in a dialysis bag and immersed in 25 mL of each of the release media under 160 rpm magnetic stirring at 37 °C. At predetermined time intervals, 1 mL aliquot of the sample was withdrawn and immediately replaced with 1 mL of fresh medium. The concentration of ND in samples was measured by HPLC (see Appendix A). The cumulative percentage of drug release was calculated and plotted versus time.

#### 2.3.5. Cell Culture

The human non-small cell lung cancer cell line A549, human osteosarcoma cell line U2OS, human embryonic kidney cell line HEK293 and human lung fibroblast line WI-38 were cultured in Dulbecco’s Modified Eagle Medium (DMEM) with 10% fetal bovine serum (FBS), 6 mM l-glutamine and 1% penicillin-1% streptomycin. Both cell lines are adherent and were propagated in a humidified incubator at 37 °C with 5% CO_2_ atmosphere. Both cell lines were a kind gift from the Penny Jeggo laboratory (GDSC, Sussex, UK).

### 2.4. Cell Proliferation/Antiproliferation/Cytotoxicity Assay

#### MTT Assay

A549 and U2OS cells were plated at a density of 7000 cells/well, and WI-38 and HEK293 were plated at a density of 6000 cells/well in 96-well plates and allowed to adhere overnight. Cells were treated with free ND or liposomal ND at the following drug concentrations: 0.5, 1, 2, 4, 8, 10, 20 µg/mL. For void liposomes, corresponding volumes to liposomal ND at the different concentrations were used as control as the concentration of lipids is the same in both formulations. After incubation in a humidified 5% CO_2_ incubator at 37 °C for 72 h, the drug-containing media was removed and MTT-containing media at a final concentration of 0.833mg/mL was added to each well and plates were incubated for 3 h, after which the media was removed and formazan crystals were solubilized in 100 µL/well of dimethylsulfoxide followed by plate shaking for 1 min. Absorbance values at 570nm were read using an automated ELISA plate reader (SPECTRO star Nano microplate reader, BMG labtech, Ortenberg, Germany). The results are represented as average percentage viability with error bars representing standard deviation. Experiments were carried out in biological triplicates.

### 2.5. Genotoxicity

#### 2.5.1. Micronucleus (MNi) Formation Assay

A549, U2OS, WI-38 and HEK293 cells were seeded at a density of 1.5 × 10^5^/well in 6-well plates containing sterilized glass coverslips. After adherence, media was replaced with drug-containing media at concentrations 0.1, 0.5 and 2 µg/mL of either ND or LND, or fresh medium for control wells. Cells were incubated with the drugs for 72 h, but after 44 h only of adding the drug, cytochalasin B was added to the drug-containing medium at a final concentration of 5 µg/mL, inhibiting cytokinesis from that point up to hour 72 within the presence of the drug. At 72 h, cells were washed with PBS, fixed for 10 min in 4% formaldehyde, and DAPI-stained for 10 min using DAPI (5 µg/mL) in PBS. Coverslips were then mounted on glass slides using fluoroshield-mounting medium and kept at 4 °C until further analysis.

Micronucleus formation, considered a biomarker of chromosome breakage, was assessed by counting the number of micronuclei observed in 50 binucleated cells (BNCs) per condition and converting that to a MNi percentage. For the high ND concentration of 2 µg/mL, where BNCs were rarer to find, due to expected cell cycle checkpoint activation, a threshold of 30 BNC cells was considered minimum cutoff. The results are represented as MNi percentage fold change normalized to the control with error bars representing standard deviation. Experiments were carried out in biological triplicates.

#### 2.5.2. γH2AX/53BP1 Immunofluorescence Foci Formation Assay

A549 and U2OS cells were seeded in 6-well plates containing sterilized glass coverslips at a density of 2 × 10^5^ cells per well, and WI-38 and HEK293 cells were seeded at a density of 1.5 × 10^5^ cells per well and allowed to adhere overnight. Cells were then treated with 0.1, 0.5, 2 µg/mL of ND or LND or untreated, for 1 h, after which drug-containing media was changed with fresh media for an additional 24 h (24-h washout). After the end of treatments and wash out periods, cells plated on glass slides were washed 2X with PBS, fixed for 10 min with 4% (*w*/*v*) paraformaldehyde and permeabilized for 3 min with 0.2% Triton-X. Cells were rinsed with PBS and incubated with anti-gH2AX (Ser139) (Merck Millipore, 05-636) and anti-53BP1 (polyclonal antibody prepared and provided as a generous gift by Dr. Raimundo Freire, hospital of the Canary Islands, Spain) diluted in PBS +2% (*w*/*v*) BSA for 1 h at room temperature. Cells were washed three times with PBS, incubated with FITC-conjugated antimouse secondary antibody (chicken antimouse secondary antibody Alexafluor488, Thermofisher scientific, A21200) and TRITC-conjugated antirabbit secondary antibody (goat antirabbit secondary antibody, Alexafluor555, Thermofisher scientific, A21428) diluted in PBS + 2% BSA for 1 h in the dark, then incubated with 4′,6-diamidino-2-phenylindole (DAPI) for 10 min and washed three times with PBS.

Slides were mounted on glass slides using fluoroshield-mounting medium and visualized using an Olympus IX70 fluorescence microscope. Fifty nuclei per condition were analyzed and scored to fall into one of three categories; nuclei with 5 γH2AX-53BP1 foci or less which was considered the basal level of DNA damage (negative), nuclei with more than 5 γH2AX-53BP1 foci (positive), and those with pan nuclear γH2AX staining. These results are demonstrated as percentages with error bars representing standard deviation. Experiments were carried out in biological triplicates.

### 2.6. Statistical Analysis

Experiments were carried out in triplicates and the mean values were calculated. The statistical significance of mean values for different conditions was assessed using two-factor ANOVA with replication, with a cut off *p* value of 0.05. Error bars represent ± standard deviation. For Anova tests of significant *p* value, multiple pairwise comparisons, post hoc comparisons, were carried out using Tukey HSD test, with again a cut off *p* value of 0.05 to identify significantly different conditions/treatments.

Statistical analysis of the in vitro release study was conducted using student *t*-test (paired), with a cut off *p* value of 0.05.

### 2.7. Uptake of Platinum by the Cell Lines

U2OS, A549 and Hek293 cell lines were seeded at a count of 0.7 × 10^6^ cells, and left overnight to adhere to the bottom of the plates. Afterwards, the old media was discarded and the cells were either supplemented with complete media (control cells), supplemented with nedaplatin each at its IC50 value, or supplemented with lioposomal nedaplatin each at its IC50 value. After 24 h, the media was collected (wash), and the cells were washed twice with PBS, detached by trypsinization and counted.

The amount of platinum was quantified with an inductively coupled plasma mass spectrometer (ICP-MS) as follows. The samples were placed in PFA advanced composite vessels and digested in a microwave (TOPwave, Analytik Jena AG, Jena, Germany) with 2 mL of high-purity HNO3 (to reach 25%) and 0.6 mL of H2O2 (to reach 10%). The microwave program for 8 vessels was 1 min at 250 W, 1 min at 0 W, 5 min at 400 W, 6 min at 600 W and 750 W at 8 min. The digested samples were evaporated to dryness in Teflon vessels. The samples were diluted with DI (deionized water) until 14 mL. All solutions were prepared with deionized water (Milli-Q-ultrapure water systems, Millipore, Watford, UK). Pt stock solution used was 1000 mg/L, (Merck, USA). The measurements were obtained by using 8800 Triple Quadrupole ICP-MS (Agilent, Santa Clara, CA, USA) [22].

## 3. Results and Discussion

### 3.1. Preparation of Stealth Liposomes Containing ND

The liposomal formulations were prepared using the thin film method as detailed elsewhere [21,23].

In this method, the water-soluble ND was passively encapsulated, while several parameters such as lipid composition, PEGylation, particle size, zeta potential, lipid to cholesterol ratio and ND to lipid ratio were optimized.

The liposomal formulation produced was shown to have 89% EE of ND, while zeta potentials of −33.50 mv and −40.70 mv (Figure 1) were obtained for the LND and the void liposomes, respectively. These negative zeta potential values are favorable for increasing liposome stability through the reduction of particle aggregation. LND and void liposomes were both shown to have a homogenous particle size distribution of around 150 nm as shown in Figure 1, while Appendix A list detailed size statistics for all particles prepared. The particle size is expected for vesicles extruded through 0.1 µm filters [21]. Figure 1 also shows a low polydispersity index (<0.1) for both LND and void liposomes indicating narrowly dispersed nanostructures with little evidence of aggregation in solution. The TEM images presented in Figure 1C also show LND particles with uniform, homogenous and spherical shapes with smooth surfaces.

### 3.2. In Vitro Drug Release

Figure 1D shows release profiles of ND and LND in PBS and FBS media at pH 7 and 6.5, respectively. FBS media more closely resembles the in vivo environment than PBS as it contains proteins, growth factors and antibodies. Moreover, the lower pH of FBS allowed us to study the release profiles in an environment that more closely resembled the usually acidic environment of cancer cells. All release profiles resulted in similar shape curves and were shown to follow the Huguchi model. This suggests a diffusional process that may be attributed to not only the diffusion of the enveloped ND within the liposomal formulation but also to a significant amount of surface immobilized ND in the formulation. Figure 1D shows the LND formulation to have a higher release rate in FBS relative to PBS, most likely due to the proteins in FBS. In PBS media, the drug release from LND was also shown to be slightly slower relative to the ND curve. Our results are in contrast to previous reports on Lipoplatin and SPI-077, being two liposomal formulations of cisplatin, showing significant increased drug retention relative to the nonformulated drugs producing significantly different release profiles [1,24,25]. The difference between the results obtained here and those previously reported may be due to the differences in drug solubilities used, as well as to the relatively high drug-to-lipid molar ratio (1:2) employed in the LND formulation compared to 1:10 and 1:70 drug to lipid molar ratios for lipoplatin and SPI-077, respectively [1,24,25]. This increased molar ratio of lipids causes the reduced mobility of drug across the liposomal membranes and thus the reported slower drug release.

### 3.3. Intracellular Accumulation of Platinum

Following the in vitro characterization of LND, we tested the intracellular accumulation of platinum in two cancer cell lines (A549 and U2OS) as well as a noncancer line (HEK293). Total platinum uptake in samples of one million treated cells accurately quantified by an inductively coupled plasma mass spectrometer (ICP-MS) directly reflected enhanced cellular uptake of the encapsulated drug (LND) in both cancer cell lines but not in the normal cell line (Figure 2A). The platinum uptake ratios (intracellular: extracellular platinum) was about two-fold higher in LND-treated U2OS cells compared to ND-treated cells and about three-fold higher in LND-treated A549 cells, while there was no significant difference in platinum uptake ratios in the normal cell line (Figure 2B). These data suggest a cancer-specific uptake enhancement of nedaplatin by liposomal encapsulation.

### 3.4. Cytotoxicity

Following the findings that LND leads to superior platinum accumulation in the cancer cell lines, we decided to test if this would translate to cytotoxicity gains. The sensitivities to nedaplatin and liposomal nedaplatin of the two cell lines A549 and U2OS as well as two noncancer lines were tested by MTT assay (Figure 3 and Appendix A). The MTT assay is a widely used quantitative and sensitive method for evaluating cell viability and cytotoxicity for screening of drugs. The cell lines differed in their sensitivity to ND with A549 being more sensitive. A549 is a non-small cell lung cancer (NSCLC) cell line, the main cancer type nedaplatin is used for treating. A large number of publications have studied the antiproliferative effect of nedaplatin in NSCLC, with a few studies investigating the expression of apoptotic markers and altered cell cycle distribution in response to nedaplatin treatment [20,26,27,28]. However, genotoxicity or DNA damage induction, assumed to be the main mechanism of action of all platinum drugs, are not assessed in any of these publications and will be addressed in this study. Moreover, this study is the first to explore the cytotoxicity of nedaplatin in the osteosarcoma cell line U2OS, which is of relevance as cisplatin is considered the standard regimen and most efficacious anticancer drug for osteosarcoma patients, where however, treatment failure is common owing to resistance [29].

Strikingly, both cell lines showed significantly higher cytotoxicity when treated with liposomal nedaplatin compared to free nedaplatin (*p* < 0.05). Importantly, the void liposome was shown not to be, by itself, cytotoxic.

Significantly, encapsulation of nedaplatin allowed for the cytotoxic effect of the drug to be exerted at a lower concentration compared to the free drug. When treated with LND, A549 cells showed a significant increase in cytotoxicity from the lowest drug concentration (0.5 µg/mL) all the way to the higher drug concentrations when compared to ND treated cells (Figure 3A). Although less sensitive than A549 cells, U2OS cells also showed a significant reduction in cell viability at a lower concentration of nedaplatin when the drug was encapsulated (Figure 3B). Taken together our results show that on two different cells lines, LND exerts a significantly greater cytotoxic effect compared to free nedaplatin. Finally, we performed a Cytokinesis Block Proliferation Index (CBPI) (Appendix A) which further demonstrated the superior antiproliferative effect of liposomal nedaplatin in both cell lines as treatment with LND showed lower proliferation index values, and a dose-dependent response across the used concentrations. CBPI indicates the average cell cycles per cell during the period of exposure to cytochalasin B in different conditions, and can be used to assess cell proliferation and the cytotoxic/cytostatic effect of a drug. Next, we tested the sensitivities to nedaplatin and liposomal nedaplatin on two noncancer lines and found that the nedaplatin cytotoxicity is not cancer-cell specific. Wi-38 cells, a human lung fibroblast line, were found to be less sensitive that A549 cells (up to 4 µg/mL) but more sensitive than U20S cells (Figure 3). The void liposome was again not toxic to the Wi-38 cells even at the highest tested concentrations. However, the added cytotoxicity observed with A549 and U20S cells when using encapsulated nedaplatin was not observed with the Wi-38 cells (ND IC50 3.07 µg/mL–LND IC50 3.67 µg/mL). This result was also observed with HEK293 cells (human embryonic kidney cells), where again no added cytotoxicity was observed when using liposomal nedaplatin (ND IC50 5.23 µg/mL–LND IC50 6.33 µg/mL). These results raise the intriguing possibility that the sensitization afforded by encapsulating nedaplatin is cancer-cell specific. Although nedaplatin is still toxic to normal cells, a cancer-specific sensitization with liposomal nedaplatin would mean that in an in vivo scenario, the same amount of cancer cell toxicity could be achieved with a lower nedaplatin concentration, with no added toxicity to noncancer cells. The improved drug uptake in cancer cell lines (Figure 2) but not normal cell lines supports and correlates with the cytotoxicity results.

### 3.5. Genotoxicity

#### 3.5.1. Micronucleus (MNi) Formation Assay

Dose-dependent genotoxic damage was observed with increasing concentrations of nedaplatin or liposomal nedaplatin assessed through micronucleus formation induction (Figure 4). Micronuclei are extranuclear bodies containing chromosomal fragments and/or whole chromosomes lagging behind in anaphase. MN assay can be used to show both clastogenic (resulting from unrepaired DSBs) and aneugenic effects (resulting from mitotic spindle damage), where usually a studied compound induces one type of MN [30,31,32,33]. For example, ionizing radiation and anthracyclins mainly induce clastogenic micronuclei, whereas vinca-alkaloids mainly induce aneugenic micronuclei. Since MN are formed in cell division and an accurate estimation of MN frequency can only be estimated in cells that have completed their first division after treatment with the studied agent, the use of Cytokinesis inhibitors is used [34]. Linear dose-micronuclei induction responses are reported in several studies. Furthermore, the presence of micronuclei is associated with apoptosis [35].

Strikingly, a significant fold change in genotoxic damage, represented in micronucleus percentage, was observed between liposomal nedaplatin and free nedaplatin, demonstrating superiority of LND in inducing chromosomal damage. This fold change of LND vs. free ND varied across drug concentrations and differed between the two cell lines (Figure 4). In U2OS cells, roughly a two-fold increase in micronucleus formation was observed across the tested drug concentration range. In the more sensitive A549 cells, a massive six-fold increase in micronucleus formation was observed at the lowest drug concentration (0.1 μg/mL) while a 3–4 fold increase was observed at the 0.5 μg/mL drug concentration. The fold-change increase in micronucleus formation at the highest drug concentration in LND vs. free ND, however, was nonsignificant in A549 cells (Figure 4). This is not surprising as with high levels of genotoxicity binucleated cells with MNi begin to decline, and further genotoxic damage does not indefinitely translate into greater numbers of micronuclei. At such high levels of genotoxic stress (resulting from drug concentrations greater than the observed IC50 in our MTT experiments), cell-cycle arrest is initiated and cells do not undergo mitosis, leading to a lower number of BNCs and/or execution of apoptosis (Appendix A). On the other hand, at the highest concentration in U2OS cells, there still is a significant increase in LND vs. ND MNi formation. This can be attributed to the resistance of U2OS as observed in MTT at that concentration (Figure 3), unlike A549 where there was significant cell death. This is consistent with a previous study comparing the anticancer effects of cisplatin and carboplatin on a panel of osteosarcoma cell lines, where U2OS has stood out as resistant to the cytotoxic effects of both tested drugs [31]. Resistance to the platinum-based drugs in that study was attributed to either more efficient repair of platinum-induced DNA damage, or the ability to evade apoptosis. This explanation is also supported by our obtained results on MTT and CBPI indices for U2OS at the tested concentrations.

Next, we tested MNi induction in our noncancer cell lines. Following exposure to either nedaplatin or liposomal nedaplatin, both WI-38 and HEK293 cells showed a mild dose-dependent increase in MNi formation, but it was a much smaller increase compared to that observed in cancer cells (Figure 4 and Appendix A). Furthermore, liposomal encapsulation of the drug did not produce a significant increase in MNi formation in these cells unlike that observed in the cancer cells (Figure 4). We attribute this difference to an intact DDR response in the noncancer cells that prevents the cells from replicating in the presence of DNA damage. Consistent with this notion, we observed a dramatic reduction in the number of binucleated cells in both noncancer lines for all tested nedaplatin concentrations. Our data suggest that in the noncancer cells, nedaplatin induces a robust DDR response that prevents cell cycle progression and initiates DNA repair. Prolonged exposure (72 h) of the cells to nedaplatin ultimately leads to cell death as indicated by our MTT data (Figure 3). Our cytotoxicity (Figure 3) and genotoxicity data (Figure 4) showing cancer-specific gains by liposomal encapsulation of nedaplatin, are consistent with our platinum accumulation data in which cellular platinum uptake is enhanced in LND treated cancer cells.

We conclude that the DNA damage caused by even the lowest tested doses of free nedaplatin is sufficient to trigger a DDR in the noncancer cells. Consequently, the additional genotoxicity afforded by liposomal encapsulation, which was mostly not significant, does not provide a significant increase in cytotoxicity in noncancer cells. In contrast however, the superior cytotoxicity of LND vs. free nedaplatin in the cancer cells, is directly linked to a more potent induction of genotoxicity and a modified DDR. The excessive DNA damage induced by LND, or even ND, met by the genomic instability hallmark of the cancer cells ultimately leads to cell death. The genomic instability of the cancer cell lines is clear in the high MN frequency percentages (Appendix A), where many counted BNC had poly-micronuclei, therefore significantly increasing the MN%.

#### 3.5.2. γH2AX Foci Analysis

In order to consolidate our findings showing the superior cytotoxicity of LND vs. free nedaplatin we used a second assay for assessing genotoxicity. Histone H2AX phosphorylation (γH2AX) is a highly sensitive marker of DNA damage and has been used extensively for the detection of DNA double stranded breaks (DSBs) [36]. Following the induction of DSBs, histone H2AX phosphorylation extends for megabases away from the DSB site leading to the formation of millions of γH2AX molecules that can be visualized by immunofluorescence as ionizing radiation-induced foci (IRIF) [36,37,38]. The formation of γH2AX triggers a complex cascade of signalling events collectively termed the DNA damage responses (DDR). Following ionizing radiation, the DDR involves cell cycle checkpoint activation and DSB repair by either nonhomologous end joining (NHEJ) or homologous recombination (HR) [35,36]. The role of γH2AX in sensing DSBs is highlighted by H2AX knockout mice that display a defective DDR and high levels of genomic instability [39,40,41].

In addition to DSB inducing agents, γH2AX has also been shown to be a marker of DNA damage induced by DNA interstrand crosslinking (ICL) agents such as platinum-based drugs and is detected with 6–10 times lower concentrations of drugs compared to detection of ICLs using the comet assay [41]. Although the role of γH2AX in sensing ICLs is not well understood, foci induced by ICL agents could reflect replication associated DNA DSBs resulting from collapsed replication forks [42]. However, it is also possible that γH2AX is a more general marker of DNA damage, not restricted to DSBs [43,44]. Here we primarily used γH2AX foci quantification to further consolidate our assessment of the genotoxic potential of LND vs. free nedaplatin (Figure 5).

To gain insight into the type of DNA damage induced by ND and LND, we also looked at 53BP1 foci formation. The 53BP1 is a mediator protein in the DNA damage response that readily forms ionizing radiation induced foci (IRIF) and has been extensively used as a marker for DSBs [45,46,47]. We postulate that areas of 53BP1 and γH2AX colocalization induced by ND and LND represent “true” DSBs resulting from collapsed replication forks (Figure 5A) as supported by previous studies [40]. Cells positive for γH2AX foci but not 53BP1 likely represent cells in G1 phase as has been reported previously for UV treated cells [43] or as postulated in another study, dividing cells with unbroken stalled forks subject to further processing or more complex structures [44].

Previous experiments using γH2AX as a DNA damage marker following exposure to platinum-based drugs have demonstrated that peak foci numbers are observed 14–24 h post drug treatment [48]. Moreover, persistence of γH2AX foci 24 h after treatment is a useful indicator for cell sensitivity/response to killing by the drug, as during this recovery period cells have the opportunity to transit S phase which is when the ICL lesions would be most toxic as they are translated to DSBs [49]. We treated the cells with three different concentrations of free or liposomal nedaplatin and then assessed the cells for γH2AX and 53BP1 foci after 24 h. Both LND and free nedaplatin induced γH2AX foci formation in a dose dependent manner as the percentage of γH2AX foci positive cells increased with increasing drug concentrations (Figure 5B). Strikingly however, across all three drug concentrations the percentage of γH2AX foci positive cells was greater in cells treated with LND vs. free nedaplatin (Figure 5B–D). Furthermore, these results could be an underestimation of the true LND performance as with increasing drug concentrations a higher percentage of cells showed pan-nuclear γH2AX staining which meant that foci could not be counted (Figure 5C–E). This was most prominent in U2OS cells treated with free or liposomal nedaplatin, as at the 0.5 μg/mL drug concentration, there was no significant difference between foci positive and negative cells but a striking three-fold difference in γH2AX pan-nuclear in favor of the encapsulated drug (Figure 5D–E). Pan-nuclear γH2AX staining is considered a preapoptotic signal and a marker of widespread replication fork collapse following extensive DNA damage. Indeed, this idea is supported by our observations during our analysis of cells with changes in nuclear morphology characteristic of apoptosis (Appendix A). A greater percentage of cells undergoing apoptosis in cells treated with LND vs. free nedaplatin would also be consistent with our cytotoxity findings (Figure 3 and Figure 4).

Next, we assessed γH2AX foci n in the noncancer cells, WI-38 and HEK293 (Figure 5F,G and Appendix A). In both cells lines we observed a smaller percentage of γH2AX positive cells (more than 5 foci) and pan-nuclear cells when compared to the cancer lines. This result is consistent with our MNi data where we also observed reduced MNi formation in the noncancer cells compared to the cancer cells. Our data suggest that even at the low drug concentrations used for our genotoxicity assays, the noncancer cells activate a robust DDR response that triggers cell cycle arrest and prevents entry to S phase which is when the ICL lesions would be most toxic as they are translated to DSBs (detectable as γH2AX foci). In contrast, cancer cells are able to bypass cell cycle arrest which leads to collapsed replication forks and the formation of DNA DSBs (Figure 5B–E). The amount of platinum accumulation is greater in cancer cells treated with liposomal nedaplatin vs. free nedaplatin (Figure 2), thus leading to a greater number of γH2AX foci which translates to the increased cytotoxicity observed by MTT (Figure 3). In the noncancer cells, the number of induced γH2AX foci is slightly higher in the cells treated with liposomal nedaplatin vs. free nedaplatin (Figure 5F) but this marginal difference does not translate to cytotoxicity differences (Figure 3).

In summary, we have used two different assays to demonstrate that LND has superior cellular accumulation and genotoxicity to free nedaplatin on two different cancer cell lines and this is a probable cause of the increased cytotoxicity observed in these two cell lines. Importantly, this appears to be cancer specific as it was not observed on the tested noncancer cells.

## 4. Discussion

The aim of this study was to evaluate the efficacy of a designed formulation of liposomal nedaplatin. The physicochemical properties of a designed liposome determine whether the liposomal drug has superior pharmacokinetic properties observed in successful cellular uptake, efficient release and selectivity towards cancer cells. Here we have synthesized a PEGylated liposomal formulation of nedaplatin with very promising characteristics, given the drug encapsulation efficiency and in vitro drug release profile (Figure 1). In order to recreate conditions closer to an in vivo scenario, we also tested our formulation in FBS. Under these conditions (presence of proteins and lower pH), our formulation maintained its promising characteristics.

Next, we tested our formulation on human cancer and noncancer cells. First, we used mass spectrophotometry for accurate quantification of platinum uptake following treatment in culture. Our data provide a-proof-of-concept that the nanoparticle designed in fact improves cellular uptake of the drug in cancer cells (Figure 2). LND thereby acts as a more potent form of the drug, potentially overcoming the problem of the resistance of cancer cells to soluble platinum drugs and their slow entry into cells as the uptake of soluble platinum compounds is influenced by factors such as ion concentration, pH, presence of reducing agents, presence of transporters or channels, while polymeric nanoparticles are taken up by cells by other processes. Importantly, the liposome-mediated drug uptake was enhanced in cancer cell lines but not in the normal cell line.

We also characterized the performance of our formulation on human cancer cells and noncancer cells in terms of cytotoxicity and genotoxicity. In our experiments an enhanced cytotoxic/antiproliferative effect of liposomal nedaplatin was selectively observed on both tested cancer cell lines (Figure 3). This is most likely due to the higher cellular drug uptake of LND, compared to ND, which was then successfully released inside the cell. Furthermore, it was indirectly deduced from MN and γH2AX/53BP1 experiments that liposomes did not interfere with the drug biodistribution inside the cell and allowed drug accumulation in the nucleus, leading to DNA platination. Since liposomal nedaplatin led to higher platinum uptake and induced greater DNA damage compared to the free drug, this also translated to greater cytotoxicity.

When using our formulations on noncancer cells we did not observe an enhanced cytotoxic or genotoxic effect of liposomal nedaplatin. We attribute this finding to the differences in the DDR between the cancer and noncancer cells. In our noncancer cells, exposure to either ND or LND triggered a DDR response that caused cell cycle arrest and limited the formation of γH2AX foci and micronuclei. Prolonged exposures (72 h) to either ND or LND did lead to cytotoxicity as measured by MTT but no additional cell killing was achieved by encapsulation of the drug.

This is the first study to show superior cellular platinum accumulation, cytotoxicity and genotoxicity in A549 and U2OS cells using PEGylated liposomal nedaplatin. Here we have provided an important first step in characterizing the mechanism of action and genotoxicity of ND by investigating the formation of ICL-associated DSBs, inferred by MN formation and directly visualized by γH2AX/53BP1 foci. We have also shown how the cytotoxicity and genotoxicity of ND can be enhanced by liposomal encapsulation. Further in vitro studies against a panel of normal and cancer human cells are needed to further assess the potential of our formulation. Of particular interest will be the performance of the formulation against rapidly dividing normal cells as well as cancer cells with inherent and acquired resistance to cisplatin. Finally, studies utilizing animal models are needed to assess whether the superior characteristics of liposomal nedaplatin observed in vitro are maintained in vivo. In cancer treatment, nedaplatin leads to dose-limiting thrombocytopenia caused by marrow suppression. Based on the results of this study showing cancer cell selectivity, we hypothesize that liposomal nedaplatin will afford cancer-cell killing at significantly lower concentrations than would have been used with the free drug, therefore decreasing the possibility of developing side effects, such as thrombocytopenia, that are observed at the therapeutic dose of the free drug. Furthermore, LND is expected to exploit the enhanced permeability and retention (EPR) effect in vivo leading to its passive targeting and accumulation in cancer tissues [12], where, due to the relatively large size of the liposome, compared to the low molecular weight of the free drug, the liposomes are not able to pass through normal tissue vasculature, but can penetrate through the leaky tumor vasculature. Together with ineffective tumor lymphatic drainage, liposomal drugs accumulate in cancer tissue in vivo therefore increasing the therapeutic potential of the encapsulated drug. Such studies are important as they can contribute towards maximizing the potential of platinum-based drugs in cancer therapeutics.

## Figures and Tables

**Figure 1 pharmaceutics-12-00863-f001:**
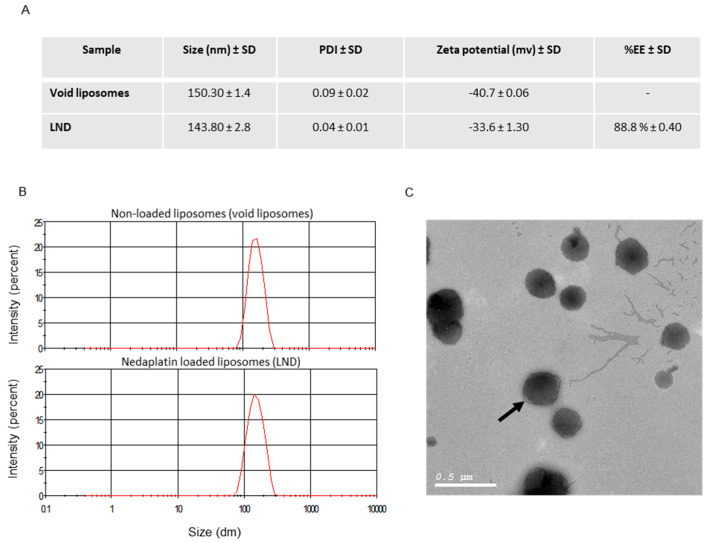
Liposome preparation and characterization. (**A**) Liposome characterization. (**B**) Size distribution plots as obtained at a scan rate of 140 counts per second at 25 °C. Top panel is for nonloaded liposomes while the bottom panel is for nedaplatin-loaded liposomes. (**C**) TEM image for LND formulation showing uniform spherical structures. The coating PEG layers are also shown as lighter outer circles (see arrow). (**D**) Nedaplatin and formulated nedaplatin release profiles in PBS and FBS media. Circles represent free nedaplatin in PBS, squares are liposomal nedaplatin in PBS and triangles represent liposomal nedaplatin in FBS.

**Figure 2 pharmaceutics-12-00863-f002:**
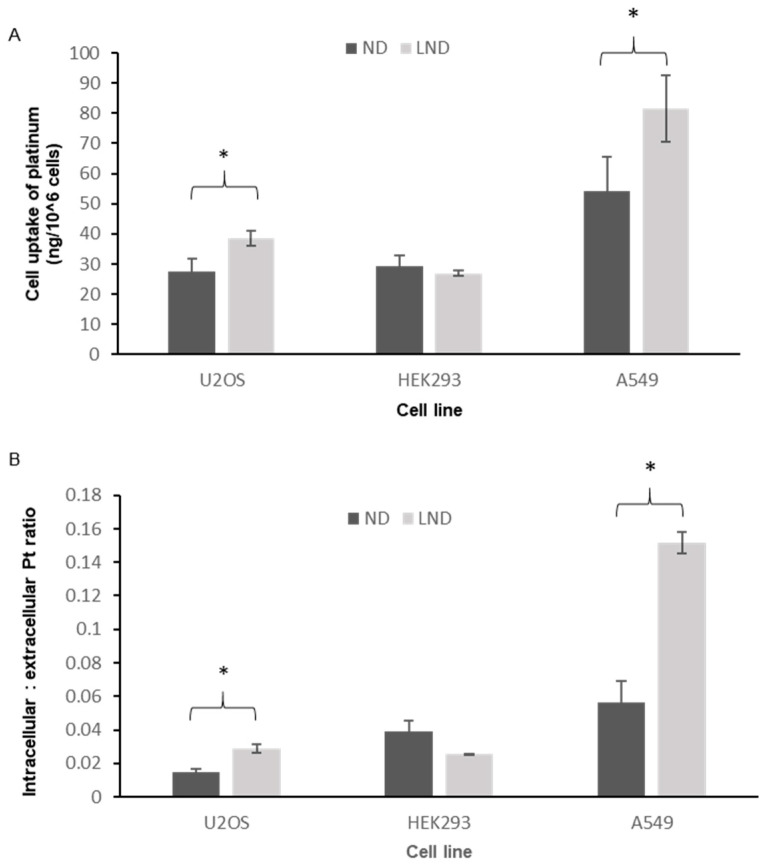
Uptake of platinum by cancer and noncancer cells. (**A**) Total platinum uptake (ng) in samples of one million treated cells accurately quantified by ICP MS. (**B**) Platinum uptake ratios (intracellular: extracellular platinum) in cancer and noncancer cells. In both cases, a statistically significant increase in cellular uptake of platinum with PEGylated liposomal nedaplatin (LND) compared to ND was observed specifically in the cancer cells lines (*p*-value < 0.05). Multiple pair-wise *t*-tests showed the cell lines where a significant difference in platinum uptake between LND and ND was observed. These points are indicated by asterisks * (*p* < 0.05) in the charts above.

**Figure 3 pharmaceutics-12-00863-f003:**
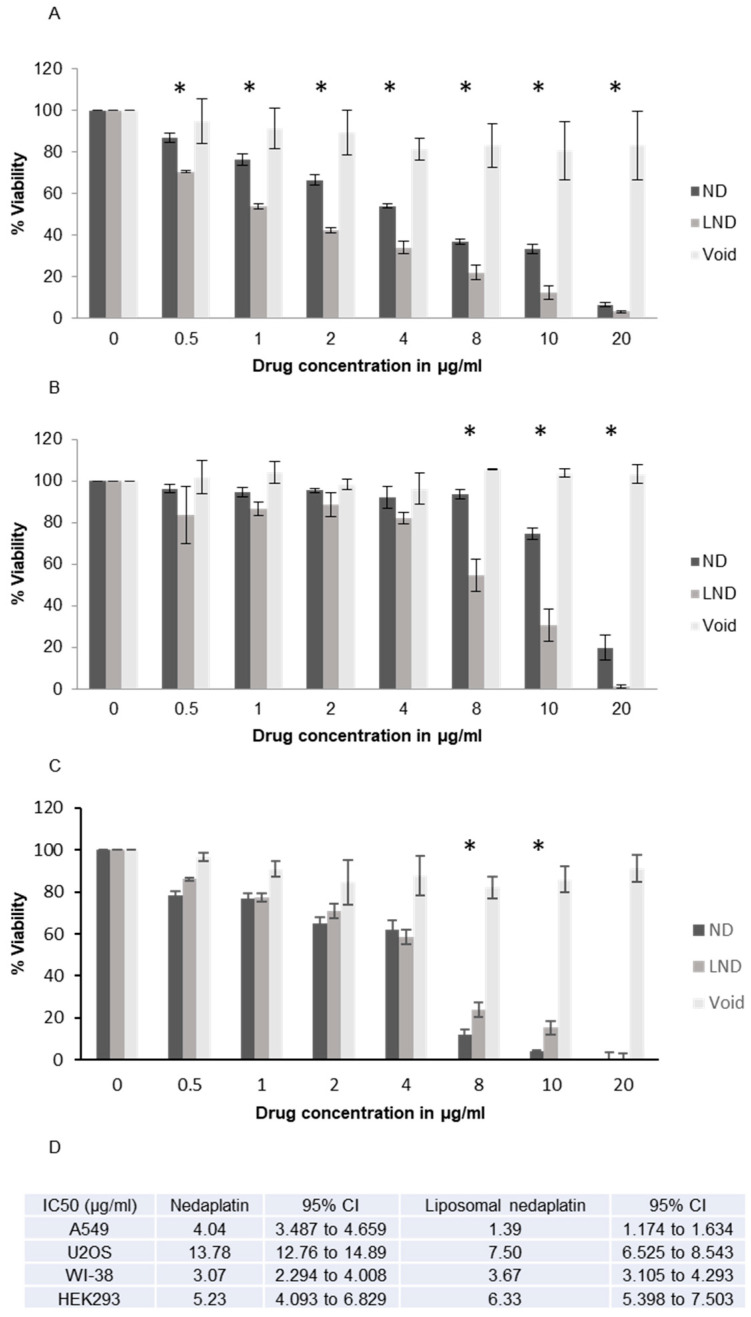
Evaluating cytotoxicity of free and liposomal nedaplatin after 72 h of drug exposure using MTT assay in (**A**) A549, (**B**) U2OS, and (**C**) WI-38. (**D**) IC50 table. An overall statistically significant difference in cell viability was observed between LND and ND in A549 and U2OS (*p*-value < 0.05) with LND being more cytotoxic. This was not observed in WI-38. Multiple pair-wise *t*-tests show the concentrations at which a significant difference between both drugs was observed. These points are indicated by asterisks * (*p* < 0.05).

**Figure 4 pharmaceutics-12-00863-f004:**
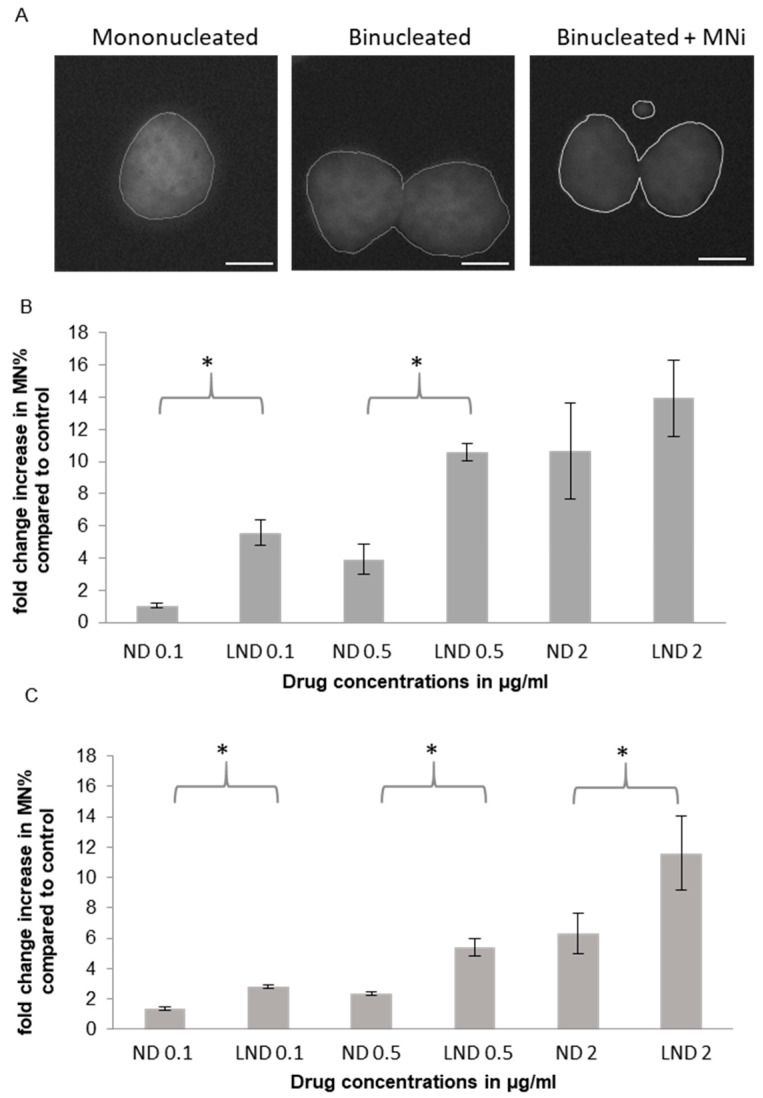
Genotoxicity of free and liposomal ND in different cell lines assessed through micronucleus formation induction. (**A**) Representative image of MNi. (**B**) A549; (**C**) U2OS and (**D**) WI-38 shows a significant increase in MNi induction between free and liposomal ND in cancer cell lines A and B, but not in normal cell lines, D. Multiple pair-wise *t*-tests show the concentrations at which a significant difference between both drugs was observed. These points are indicated by asterisks * (*p* < 0.05). Scale bars are 5 µm.

**Figure 5 pharmaceutics-12-00863-f005:**
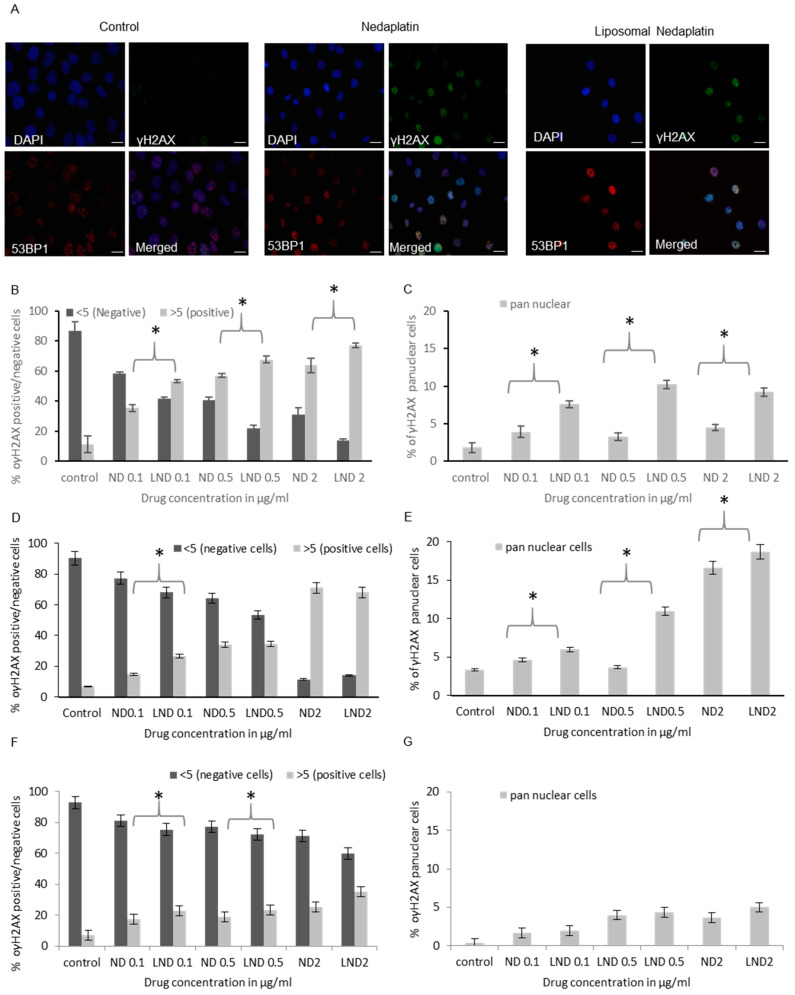
Genotoxicity of free and liposomal ND in cancer and noncancer cell lines assessed through γH2AX induction. (**A**) Representative immunofluorescence images of control, ND and LND treated A549 cells. Quantification of γH2AX positive and γH2AX pan nuclear cells, respectively, in (**B**,**C**) A549, (**D**,**E**) U2OS, (**F**,**G**) WI-38 cells. While an overall statistically significant increase in the percentage of cells positive for DNA damage observed with LND compared to ND was shown in all cell lines, a statistically significant increase in the γH2AX pan-nuclear signal was only observed with LND in the cancer cell lines (*p*-value < 0.05). Multiple pair-wise *t*-tests showed the concentrations of treatment where a significant difference between LND and ND DNA damage is observed. These points are indicated by asterisks * (*p* < 0.05) in the charts above. Scale bars are 10 µm.

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
