# Peer review of "Encapsulation of Nedaplatin in Novel PEGylated Liposomes Increases Its Cytotoxicity and Genotoxicity against A549 and U2OS Human Cancer Cells"

_pharmaceutics, 2020, doi:10.3390/pharmaceutics12090863_

Round 1

Reviewer 1 Report

The manuscript by El-Shafei et al. presents results on physicochemical properties and enhanced antiproliferative activity of newly formulated PEGylated liposomal nedaplatin. Authors provide evidence that liposomal nedaplatin has increased antitumor activity compared to nedaplatin against human A549 and U2OS cancer cell lines. Overall, the findings are interesting.

Recommendations:

Lanes 186 and 191: Subheadings “2.5.2. H2AX/53BP1 foci formation assay” and “2.5.3. Immunofluorescence” might be merged.

Lane 226: Correct a zeta potential of the void liposomes to “-40.70 mV”.

Figure 1B: Top and bottom graphs of particle size distribution should be directly labelled as non-loaded liposomes (void liposomes) and nedaplatin-loaded liposomes (LND).

Lane 257: Legend to the figure “Circles represent liposomal nedaplatin in PBS, squares are free nedaplatin in PBS and triangles represent liposomal nedaplatin in FBS” are mixed up.  Correct the labels or the legend accordingly.

Lane 296: References for Cytokinesis Block Proliferation Index assay and its use to determine cytotoxicity of a test compound has to be cited (e.g. PMID: 18602494).

Figure 3B and 3C: Correct the legend for nedaplatin (ND) and liposomal nedaplatin (LND) (accordingly as in the Figure 4B-D).

Lane 370, Figure 4: Correct the legend to the figure. It should state “Figure 4. Genotoxicity of nedaplatin and liposomal nedaplatin in A549 and U20S cells assessed through γH2AX foci induction.

Lane 371: Authors state: A. Representative immunofluorescence images of control, ND and LND treated cells. Images of which cells are presented here? A549 or U2OS?

Lane 382: Include the cited paper of Olive and Banath, 2009 in the Reference list.

Lane 396: Supplemental Figure 3 – indicate the concentrations of ND and LND used to induce apoptosis of A549 cells.

Reviewer 2 Report

This paper reported nedaplatin encapsulation liposome with basic in vitro analyses. Although authors succeeded in showing that nedaplatin liposome outperformed, overall benefit of this paper is low. Preparation method of liposome is not new, and this study lacks in vivo study. Besides these issues, following issues should be addressed.

1) Mechanisms underlying improved outcome of liposome over free drug is unclear. In discussion, authors mentioned that the mechanism may be due to increased cellular uptake. Nanomedicine can improve the efficiency of anti-cancer drug in other mechanisms, such as the change in intracellular processing. For example, liposome is presumed to internalize cells via endocytosis, which can bring drugs near nucleus. This is beneficial for drug accumulation in the nucleus and for escaping from drug excretion by transporters located in cell membrane.

2) In introduction, authors describe that nedaplatin leads to thrombocytopenia caused by marrow suppression. How does liposomal formulation they developed avoid this issue?

3) Figure 3b, c, fold change per control was shown. Instead, better to describe percentage of cells possessing MN. 

4) In figure 5b,d, why did authors show both positive and negative cells? Only one of them seems enough. In addition, the total of positive and negative cells seems not to be 100%. Is this due to cells with 5 foci. 

Is the vertical axis "/negative cells" correct? "/total cells" seems better.

Reviewer 3 Report

The paper submitted for publication in “Pharmaceutics” by El-Shafeii and collaborators is related to the evaluation of the cytotoxic and genotoxic effects of PEGylated liposomal of nedaliplatin in A549 human non small cell lung cancer and U2OS human osteosarcoma cells.

The preliminary results are promising and might be of interest for a broad readership. However, before warranting publication in “Pharmaceutics” several points/issues have to be addressed.

  • When screening antitumor drugs for cytotoxicity, their genotoxic activity must be taken into consideration, to prevent secondary tumors after treatment. As stated by the authors “This is the first study to show superior cytotoxicity and genotoxicity in tumor cells” . It is not clear how the results can be applied into preclinical practice, without a concomitant evaluation of the genotoxic potential in normal cells. Therefore, it is mandatory to compare the MN frequency in non tumor cells, such as normal phenotype cells of lung and bone origin in order to verify a preferential genotoxicity versus tumor cells. These results will provide information concerning the potential risk of late-life morbidities in cancer survivors exposed to this kind of chemotherapy.
  • Besides the choice of cellular system also the treatment duration and concentration may significantly influence the MN assay outcome, mostly in the case of highly cytotoxic compounds. In order to obtain a more reliable indication about the genotoxic potential of the tested species MN frequency has to be assessed after shorter incubation times than 72 h.
  • Based on their results, the authors conclude that PEGylated liposomal nedaplatin allows a higher cellular and nuclear accumulation of ND. But this statement need to be confirmed by cellular uptake experiments that by AAS technique allow the detection of Pt amount inside the cells following treatment. Moreover, in order to confirm the results on DNA damage, DNA platination, again via AAS technique are needed.

Round 2

Reviewer 2 Report

Authors provided reasonable answers to most of the questions. There still remains several issues.

1) New data explaining the difference in cytotoxicity of liposome between cancer and normal cells is interesting. Could authors discuss why liposome and free drugs show comparable toxicity to normal cells, while liposome outperformed in the activity against cancer cells?

2) "As different cell lines have different background frequency of micronuclei, it is important to compare micronucleus induction due to treatment normalized to controls, as opposed to comparing changes in number of micronuclei/100 or 1000 Binucleated cells."

Still in this case, it would be helpful to provide percentage in total cells. Otherwise, readers cannot understand if this is a meaningful event.

Reviewer 3 Report

In response to the reviewer suggestion, the Authors provided new results on cytotoxicity and genotoxicity of ND and PEGylated liposomal nedaliplatin in normal cells. Actually, these results are interesting in supporting the therapeutic potential of PEGylated liposomal nedaliplatin, but the authors have to discuss them providing possible explanations about the selectivity against cancer cells. About that, studies aimed to detect cellular platinum accumulation in normal and in cancer cells are mandatory.  

Round 3

Reviewer 2 Report

Now, manuscript is acceptable.

Reviewer 3 Report

Unfortunately, the authors haven’t addressed the issues raised. As I already stated in the first and in the second round of the revision, the authors conclusions about the different cellular accumulation  of ND and PEGylated liposomal ND are too speculative. Experiments aimed at quantifying cellular platinum content are mandatory. The manuscript contains observations, but is not a full study.
